# KALMAN FILTER ENHANCED GROUP RELATIVE POLICY OPTIMIZATION FOR LANGUAGE MODEL REASONING

## ABSTRACT

The advantage function is a central concept in RL that helps reduce variance in policy gradient estimates. Recently, for language modeling, Group Relative Policy Optimization (GRPO) was proposed to compute the advantage for each output by subtracting the mean reward, as the baseline, for all outputs in the group. However, it can lead to high variance when the reward advantage is inaccurately predicted. In this work, we propose Kalman Filter Enhanced Group Relative Policy Optimization (KRPO) model, by using lightweight Kalman filtering to dynamically estimate the latent reward baseline and uncertainty. This filtering technique replaces the naive group mean, enabling more adaptive advantage normalization. Our method does not require additional learned parameters over GRPO. This approach offers a simple yet effective way to incorporate multiple outputs of GRPO into advantage estimation, improving policy optimization in settings where highly dynamic reward signals are difficult to model for language models. Through the accuracies and rewards obtained from math question answering and reasoning, we show that using a more adaptive advantage estimation model, KRPO can improve the stability and performance of GRPO. The code is available at https://anonymous.4open.science/r/KRPO-E1D3.

## 1 INTRODUCTION

In reinforcement learning algorithms that directly optimize the policy via gradient estimates, including Policy Gradient (PG) (Sutton et al., 1999), Actor-Critic (AC) (Mnih et al., 2016) and Proximal Policy Optimization (PPO) (Schulman et al., 2017), variance reduction of policy gradient estimate is critical for stable and sample-efficient training. A common approach to estimate the advantage of actions is to subtract a baseline (and divide by a variance), which can be interpreted as the expectation of returns by taking all sampled actions for the current state, from the observed returns. By doing so, the model is ideally expected to approximate the expected value of the return under the current policy. In many prevalent algorithms, such as AC or PPO, the baseline is often computed via an additional maintained value model. However, the value model introduces further computational costs, especially for large model training.

To overcome this issue, Group Relative Policy Optimization (GRPO) (Shao et al., 2024) is proposed to simplify the prediction of reward baselines by taking the mean reward over a group of trajectories. However, this approach can lead to high variance if an inaccurate advantage is estimated when the reward is highly nonstationary. This may lead to unstable updates and slower convergence, especially in settings where the reward signal evolves rapidly with high variance over time.

To address this issue, we propose a dynamic advantage estimation method based on a one-dimensional Kalman filter, named as **K**alman Filter Enhanced Group **R**elative **P**olicy **O**ptimization (KRPO). Instead of relying on the group mean, we treat the observed rewards as noisy measurements of a latent reward baseline signal and use the Kalman filter to model the underlying reward and noise trajectories. The resulting filtered baselines and uncertainties adapt over time, smoothing out noise while tracking underlying reward baseline values in the environment. This enables more accurate and robust advantage estimation with minimal computational overhead. Our method integrates naturally with the existing GRPO algorithm and does not require modification of policy networks.

Provides a principled way to incorporate uncertainty into advantage estimation without introducing additional learning components or tuning objectives. We evaluate our approach by comparing it to the standard group-mean approach and show how it improves training stability. In summary, this paper makes the following contributions:

- We introduce a lightweight Kalman filter approach for accurate advantage estimation. It can be seamlessly incorporated into the Group Relative Policy Optimization (GRPO) algorithm without introducing extra learning parameters.

- We empirically demonstrate that the proposed KRPO algorithm can receive higher rewards than its standard group-mean counterparts across different datasets.

## 2 KALMAN FILTER ENHANCED GROUP RELATIVE POLICY OPTIMIZATION

### 2.1 PROBLEM SETTING

We consider reinforcement learning in which an agent interacts with an environment modeled as a Markov Decision Process (MDP). In a timestep, the agent observes a state $s$, selects an action $a$ according to a policy $\pi(a \mid s)$, and receives a reward $r$. The goal is to maximize the expected cumulative return. Policy gradient methods rely on estimating the advantages to reduce the variance of gradient updates. The Group Relative Policy Optimization (GRPO) framework is applied to large language models by sampling multiple responses (i.e., action trajectories) from the policy given the same input prompt. A shared baseline is computed as the average return across all sampled responses for the same prompt, and the advantage for each output is calculated by subtracting the group sample mean. Although this reduces variance, the group mean may result in inaccurate estimates of action advantages, particularly when the rewards are very noisy. This can lead to unstable or suboptimal updates during training.

### 2.2 KALMAN-FILTERED ADVANTAGE ESTIMATION

We propose replacing the naive group mean baseline with a Kalman filter that maintains an online estimate of the expected reward and its variance to obtain a more accurate estimate of the advantages. Each observed return $r_i$ is treated as a noisy measurement of the latent expected reward signal $\hat{x}_i$, which we cannot obtain directly. The subscript $i$ refers to the index of the i-th sampled observation.

The Kalman filter maintains a posterior estimate of the reward baseline $\hat{x}_{i|i}$ and its associated uncertainty $P_{i|i}$, both updated recursively. Here, the subscripts follow the standard Kalman notation: $\hat{x}_{i|i-1}$ denotes the prior estimate of "genuine" reward baseline $\hat{x}_i$ given observations up to step $i-1$, while $\hat{x}_{i|i}$ is the posterior estimate after incorporating the observation $r_i$. Similarly, $P_{i|i-1}$ and $P_{i|i}$ denote the corresponding uncertainty (variance) estimates before and after observing $r_i$.

The update consists of two stages: (1) Prediction step and (2) Update step. For the prediction step:

$$\hat{x}_{i|i-1} = \hat{x}_{i-1|i-1}, \tag{1}$$
$$P_{i|i-1} = P_{i-1|i-1} + Q, \tag{2}$$

For the update step:

$$K_i = \frac{P_{i|i-1}}{P_{i|i-1} + R}, \tag{3}$$
$$\hat{x}_{i|i} = \hat{x}_{i|i-1} + K_i(r_i - \hat{x}_{i|i-1}), \tag{4}$$
$$P_{i|i} = (1 - K_i)P_{i|i-1}, \tag{5}$$

where $Q$ is the process noise variance and $R$ is the measurement noise variance. These are hyperparameters that control the smoothness and adaptability of the estimated baseline.

Given the posterior mean $\hat{x}_{i|i}$ and the posterior variance estimate after seeing each reward $P_{i|i}$, we define the advantage at timestep $i$ as

$$A_i = \frac{r_i - \hat{x}_{i|i}}{\sqrt{P_{i|i}} + \varepsilon},$$ (6)

where $\varepsilon$ is a small constant for numerical computation stability. This formulation has two benefits: it adaptively centers the reward using a filtered baseline, and it normalizes the advantage by the estimated uncertainty, which helps stabilize the scale of policy updates and get a more reasonable advantage estimate.

The obtained advantage estimate $A_i$ can be directly used in place of existing advantage estimates for models within the policy gradient family. In our implementation, we integrate it with GRPO by replacing the typical advantage computation with the Kalman-filtered version described above. This method does not require training a separate value network for advantage estimation, nor does it add new optimization objectives. The filter state is updated online during each policy update step, and incurs negligible additional computational cost.

## 2.3 KRPO ALGORITHM

The KRPO procedure is fully differentiable and can be implemented efficiently upon the GRPO frameworks. After getting the group rewards, instead of receiving the mean value from the group, we fetch the Kalman filtered baseline and estimated uncertainty $P_{i|i}$. By taking into account the posterior variance estimation $P_{i|i}$, the advantage can be more adaptively modeled.

---

**Algorithm 1:** Kalman Filter Enhanced Group Relative Policy Optimization (KRPO)

---

**Input:** Reward function $\mathcal{R}$, LLM policy $\pi_\theta$, reference policy $\pi_{\text{ref}}$, prompt dataset $\mathcal{D}$
**Hyperparameters:** KL weight $\beta$, group size $n$, batch size $B$, learning rate $\eta$, clipping $\epsilon$,
 process noise $Q$, measurement noise $R$ Initialize replay buffer $\mathcal{B}$, optimizer Adam$(\theta, \eta)$ ;
**for** *each training step* **do**
    **for** *each prompt* $(x, y^*) \sim \mathcal{D}$ **do**
        **for** $i = 1$ **to** $n$ **do**
            Sample trajectory $y_i \sim \pi_\theta(\cdot|x)$ ;
            Compute reward $r_i = \mathcal{R}(y_i, y^*)$ ;
        Use a Kalman Filter to estimate reward baseline $\hat{x}_{i|i}$ and variance $P_{i|i}$ from $\{r_i\}_{i=1}^n$ ;
        Compute advantage: $A_i = \frac{r_i - \hat{x}_{i|i}}{\sqrt{P_{i|i}} + \varepsilon}$ ;
        Evaluate log-probs: $\log \pi_\theta(y_i|x), \log \pi_{\text{ref}}(y_i|x)$ ;
        Compute KL: $D_{\text{KL}} = \log \pi_\theta - \log \pi_{\text{ref}}$ ;
        Store $(x, y_i, A_i, D_{\text{KL}})$ into buffer $\mathcal{B}$ ;
    **for** *each minibatch in* $\mathcal{B}$ **do**
        Compute policy loss with clipping:

$$\mathcal{L} = -\hat{\mathbb{E}}\left[\min\left(\frac{\pi_\theta}{\pi_{\theta_{\text{old}}}} A, \text{clip}\left(\frac{\pi_\theta}{\pi_{\theta_{\text{old}}}}, 1 - \epsilon, 1 + \epsilon\right) A\right) + \alpha D_{\text{KL}}\right]$$

        Perform gradient descent on $\mathcal{L}$ with gradient clipping ;

---

Worth noting, in the model training, the observations are completely randomly sampled in each iteration. Moreover, the order of samples in our implementation is arbitrary (e.g., they can be shuffled before processing), so no unintended temporal dependence is imposed on the data. It simply provides a recursive estimator with an accompanying measure of uncertainty. The ordering will thus not affect the policy optimization.

Besides, each Kalman filter operation is applied to groups of rollouts. In the KRPO setting, although the reward values for individual samples are discrete, the grouped reward observations, which aggregate multiple rollouts, are not sparse and are much more continuous in distribution. According to the Central Limit Theorem (CLT, 2008), the sum of i.i.d. sampled rewards (with finite mean and finite variance) tends to follow a Gaussian distribution. This is consistent with the assumptions of our KRPO setting.

## 3 CONNECTIONS OF GROUP-MEAN AND KALMAN FILTERING

The baseline computed in the GRPO model is the empirical mean of observed rewards. Specifically, for a group of returns $\{r_i\}_{i=1}^n$, the baseline is defined as

$$\bar{r} = \frac{1}{n} \sum_{i=1}^n r_i. \tag{7}$$

This empirical mean can be interpreted as a special case of a Kalman filter with the following assumptions: (1) The process noise variance $Q \to 0$, indicating that the latent state (expected reward) is assumed to be constant; (2) The measurement noise variance $R$ is a constant; (3) initial state is unknown, modeled with large uncertainty.

Specifically, consider the Kalman filter update equations as aforementioned Eq. 1 to Eq. 5. When $Q$ is constant and the system state does not change over time (i.e., the state transition matrix is an identity matrix and the process noise is zero), the Kalman filter update reduces to a weighted average of the observations, where the weights are determined by the prior variance and the observation noise variance. However, in practice, when the full batch of $\{r_i\}$ is available, computing the sample mean corresponds to performing a batch update using all data equally. This is equivalent to a Bayesian update with all measurements weighted equally, which can be seen as a limiting form of the Kalman filter without temporal dynamics.

This observation motivates the use of a Kalman filter, which accounts for both observation noise ($R > 0$) and possible slight changes in the latent reward ($Q > 0$), and dynamically adjusts the influence of new observations via the gain $K_i$. This leads to a more adaptive estimate of the reward advantage, making the Kalman filter a more general formulation compared to a simple average.

## 4 EXPERIMENTS

### 4.1 IMPLEMENTATION DETAILS

We adopt four datasets in our experiments, including Arithmetic (Open-Thought, 2024), OpenMath-Instruct (Toshniwal et al., 2024), MATH500 (Lightman et al., 2023), and AIME (Veeraboina, 2024). We keep the standard training/testing 80%/20% split ratio for each of the four datasets without model selection. Arithmetic and OpenMathInstruct are divided into three sub-datasets individually: *easy*, *normal*, and *hard*. More details about the datasets used in the paper can be found in the appendix. In the paper, we focus on math tasks. These tasks are suitable for experiments as they require the model to have adequate reasoning ability for resolving difficult math questions (even not easy for humans), and language model reasoning is the main focus of our KRPO paper. Moreover, math tasks have well-defined answers for reward design or introduce extra reward models, which is not the main emphasis of our paper.

We adopt a group size of 12 for both the GRPO and our proposed KRPO models. In our implementation, the Kalman filter state is reset for each prompt to avoid cross-prompt information leakage and ensure statistical independence across prompts. The training batch size is set to 16. The model is optimized using the Adam optimizer with an initial learning rate of $5 \times 10^{-6}$. To make the computation manageable, we adopt `Llama-3.2-1B-Instruct`, `Qwen2.5-0.5B-Instruct` and `Qwen2.5-1.5B-Instruct` as our base models, to individually train with the GRPO algorithm and the proposed KRPO algorithm. By default, the Kullback–Leibler (KL) divergence term is weighted by a factor $\alpha$ of 0.01 during training; variance of the filter process noise $Q$ is set to 1e-5 and the variance of the filter measurement noise $R$ is set to 1e-2. For the Arithmetic dataset, the model is trained for 16,000 steps. For the OpenMathInstruct dataset, training is conducted for 5,500 steps. For both MATH500 and AIME datasets, we train the model for 10,000 steps. For the reward design, following the DeepSeek-R1-Zero model (Guo et al., 2025) to keep simplicity, we do not use a reward model and instead assign rewards as follows: if an answer is exactly equal to the ground-truth answer, the reward is 1.0; if a ground-truth answer is contained within a predicted answer, a 0.5 reward is assigned; otherwise, no reward will be returned. We also encourage the model to think reasonably through "`<think></think>`" tags. The random seed of experiments is set to 42. All experiments are conducted on one A6000 Nvidia GPU with 48GB of graphics memory. The

proposed KRPO model and its compared models are trained and tested under the same setting to keep fair comparisons.

## 4.2 EXPERIMENTAL RESULTS

### 4.2.1 OVERALL MODEL PERFORMANCE

Table 1: Model accuracy comparison between the KRPO and other models on the Arithmetic and OpenMath-Instruct datasets. The best performance for each block with the same setting is **bolded**. The p-value (last column of table1) is calculated from the answer accuracies by one-tailed paired t-tests between each method against KRPO.

| Dataset | Models | Easy | Normal | Hard | p-value |
|---|---|---|---|---|---|
| Arithmetic | Llama3.2 (Touvron et al., 2023) | 22.516% | 7.940% | 5.076% | 3.6e-2 |
| | PPO (Schulman et al., 2017) | 64.425% | 43.684% | 17.650% | 2.3e-2 |
| | GRPO (Shao et al., 2024) | 69.468% | 45.228% | 18.599% | 4.2e-2 |
| | KRPO (Ours) | **71.764%** | **50.589%** | **20.881%** | - |
| | Improvement | **+2.296%** | **+5.361%** | **+2.282%** | - |
| OpenMath | Llama3.2 (Touvron et al., 2023) | 13.056% | 12.423% | 9.717% | 9.0e-3 |
| | PPO (Schulman et al., 2017) | 66.871% | 39.783% | 27.983% | 1.5e-2 |
| | GRPO (Shao et al., 2024) | 76.399% | 54.144% | 33.983% | 4.7e-2 |
| | KRPO (Ours) | **81.022%** | **66.517%** | **51.859%** | - |
| | Improvement | **+4.623%** | **+12.373%** | **+17.876%** | - |

We compare the accuracy performance of our proposed KRPO against several baseline models, including the original Llama3.2 model (in Tab. 1, Llama-3.2-1B-Instruct is used as the base model), Proximal Policy Optimization (PPO), and Group Relative Policy Optimization (GRPO). We calculated the accuracy by also assigning half correctness if the ground-truth answer is contained in the returned strings, e.g., ground-truth is "59", but "The answer is 59" is considered half correct. Tab. 1 shows the results on two mathematical reasoning benchmarks: Arithmetic and OpenMath-Instruct. The results are reported across three difficulty levels: Easy, Normal, and Hard.

Across both datasets and all difficulty levels, KRPO consistently outperforms baseline models. On the Arithmetic dataset, KRPO improves over GRPO by 2.296%, 5.361%, and 2.282% on Easy, Normal, and Hard subsets, respectively. Similarly, on the OpenMath-Instruct dataset, KRPO achieves substantial gains, with improvements of 4.623%, 12.373%, and 17.876% over GRPO. These improvements support that using a Kalman Filter provides a more stable and accurate advantage estimation (the only difference with GRPO), which benefits policy learning, especially on more challenging problems.

Beyond the overall performance improvements, we observe interesting trends across the two datasets. We find that the raw pretrained Llama3.2 model achieves higher accuracy on Arithmetic-Easy (22.516%) compared to OpenMath-Easy (13.056%), likely due to the short and highly patterned nature of Arithmetic questions, which allows the language model to leverage surface-level cues. However, after applying reinforcement learning-based reasoning training, the KRPO model trained on OpenMath outperforms the accuracy on Arithmetic. This highlights that Arithmetic problems, although syntactically simpler, actually require strong numerical reasoning ability, which is the weak point of the Llama3.2 models. In contrast, OpenMath problems are longer and more linguistically complex, but once the reasoning structure is learned, the model can generalize more effectively across examples.

We also show the statistical significance (p-value) of performance gaps between each model and the proposed KRPO model. All differences are statistically significant with p-values below the threshold 0.05, indicating that the observed improvements of KRPO to other models are large. These results collectively indicate that incorporating a Kalman Filter model for advantage estimation is especially beneficial in high-variance, complex reasoning environments. This aligns with the goal of improving the stability of the advantage computation during training. More performance comparisons with other models, including Direct Preference Optimization (DPO) (Rafailov et al., 2023) and fixed baseline models, can be found in the appendix.

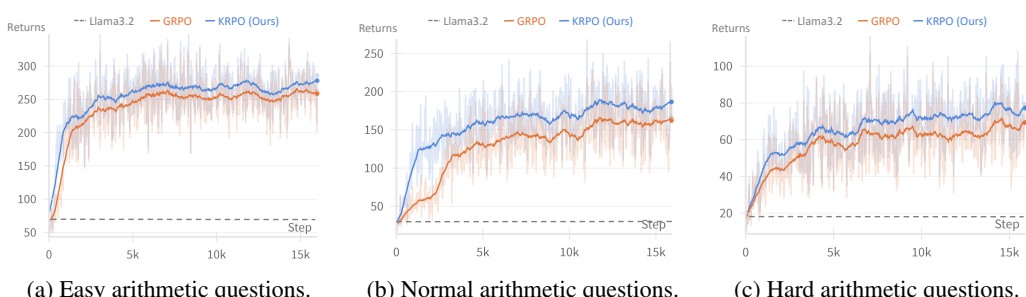

(a) Easy arithmetic questions.    (b) Normal arithmetic questions.    (c) Hard arithmetic questions.

Figure 1: The curves of returns/rewards within a batch for different difficulty levels of questions within the Arithmetic Dataset.

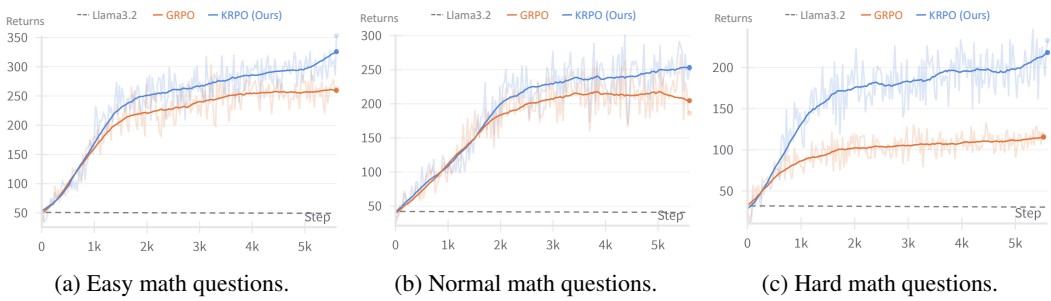

(a) Easy math questions.    (b) Normal math questions.    (c) Hard math questions.

Figure 2: The curves of returns/rewards within a batch for different difficulty levels of questions within OpenMath-Instruct Dataset.

### 4.2.2 TRAINING REWARD CURVE COMPARISON BETWEEN KRPO AND GRPO

We also show the comparison of the training return curves between KRPO and GRPO on the Arithmetic Dataset as Fig. 1 and on the OpenMath-Instruct Dataset as Fig. 2. We use a running average for reward smoothing to improve visualization quality. We set the group size to 12 and the rollout batch size to 32. In this case, the maximum possible sum of rewards at each point is 384.

On both datasets and across all difficulties, the proposed KRPO algorithm can consistently converge faster and eventually reach much higher rewards. Interestingly, on the OpenMath-Instruct dataset, the performance gap between KRPO and GRPO becomes larger as problem difficulty increases. This trend suggests that KRPO provides more stable and accurate advantage estimates in harder reasoning scenarios, where the variance in reward signals is likely higher and a naive baseline, such as the mean reward, becomes less effective. This trend is also applied to the Arithmetic easy and normal subsets, but is not observed on the Arithmetic hard dataset. It is possibly caused by the difference in the arithmetic hard data distribution from other difficulty sets.

We also train KRPO on two additional base models as shown in Fig. 3a: Qwen2.5-0.5B-Instruct and Qwen2.5-1.5B-Instruct on Normal level difficulty Arithmetic questions. The results are very similar to those on Llama-3.2-1B-Instruct, with KRPO also outperforming GRPO under the exact same experimental conditions. We've also found that the Qwen2.5 (even the smallest Qwen2.5-0.5B-Instruct) models can perform much better results than Llama-3.2-1B-Instruct.

Additionally, we conduct experiments on additional AIME and AMC data (with Llama3.2-1B-Instruct as the base model) as shown in Fig. 3b. Despite these two datasets being much more challenging for the Llama3.2-1B model than Arithmetic and OpenMath-Instruct, the results showed similar relative comparisons between KRPO and GRPO under the exactly same conditions: KRPO outperforms GRPO baseline consistently over training steps. More detailed quantitative model performance comparisons on different base models and datasets can be found in the appendix.

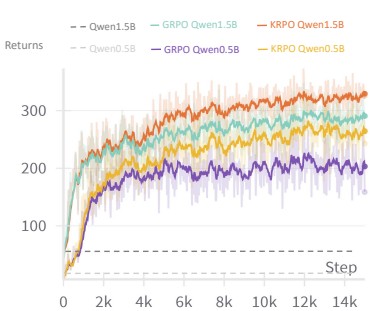 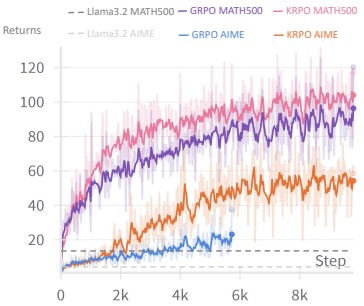

(a) Qwen-0.5B and Qwen-1.5B as base models.

(b) More results on MATH500 and AIME datasets.

Figure 3: The training return curves of additional (a) base models and (b) datasets.

## 4.3 ANALYSIS

In this sub-section, we analyze the effect of different hyper-parameters and observed phenomena (time efficiency analysis can be found in the appendix). For all experiments in the analysis section are conducted on Arithmetic data.

### 4.3.1 KL LOSS WEIGHTS $\alpha$

We visualize the training curves with different KL weights in the loss $\alpha$ of KRPO by setting it to $\{0, 0.001, 0.01, 0.05\}$ as Fig. 4a. We observe that with the 0.01 $\alpha$, it reaches the fastest convergence, as well as the highest returns. However, with 0.001 settings, the rewards are converged more slowly but eventually reach the same point; with 0.05 and 0 settings, the rewards are much worse than 0.01. So $\alpha = 0.01$ is set as the default setting for KRPO.

### 4.3.2 MODEL CONVERGENCE AGAINST PROCESS NOISE AND MEASUREMENT NOISE

We investigate the model convergence against process noise $Q$ and measurement noise $R$ as Fig. 4b. We noticed that: (1) Robustness to hyperparameter variations: When varying Q from 1e-5 to 1e-3, and $R$ from 1e-2 to 1e-1, the model converges to similar final performance, though a smaller $R$ leads to slower convergence. (2) Sensitivity to overly small $R$: When $R$ is reduced to 1e-3, the filter becomes overly confident in noisy observations, which results in unstable or poor learning signals and degrades the final performance. These results suggest that KRPO is robust to a range of Q and $R$ values, as long as $R$ is not set unrealistically low. A moderate level of measurement uncertainty helps stabilize the advantage estimation.

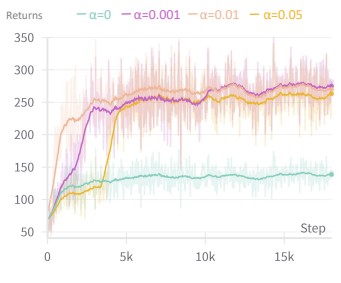 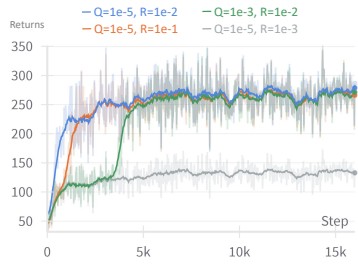

(a) KL divergence loss weights.

(b) Process and measure noises.

Figure 4: The training return curves of different (a) KL divergence loss weights and (b) process and measure noises.

Table 2: Case study for the thinking process of GRPO and the proposed KRPO model.

| Question | {"type": "Algebra", "question": "If $7^{4x} = 343$, what is the value of $7^{4x-3}$", "expected_answer": "1"} |
|---|---|
| **Model** | **Thinking Process** |
| GRPO | ✗ First, let's rewrite $7^{4x} = 343$ as $7^{4x} = 7^3$. This is because $343 = 7^3$. By equating the exponents, we get $4x = 3$. Solving for $x$, we find that $x = \frac{3}{4}$. Therefore, $7^{4 \cdot \frac{3}{4}} = 7^3 = 343$. The answer to this problem is $343$. |
| KRPO | ✓ To solve the given equation $7^{4x} = 343$, we first express both sides with the same base. Since $343 = 7^3$, we can rewrite the equation as $7^{4x} = 7^3$. This implies $4x = 3$, so $x = \frac{3}{4}$. Now that we have the value of $x$, we can find $7^{4x-3}$ by substituting $x = \frac{3}{4}$ into the expression: $7^{4\left(\frac{3}{4}\right)-3} = 7^{3-3} = 7^0 = 1$. |

From all the above experiments, we can learn: the reward characteristics vary across datasets and base models, e.g., easier tasks or stronger base models tend to produce more stable and low-variance rewards, while harder tasks or weaker base models often result in the other way around. KRPO is beneficial in settings where reward signals are noisy, as the Kalman filter provides an adaptive prediction of advantages. However, KRPO may become less effective if the Kalman filter's parameters (process and measurement noise) are not aligned with the reward dynamics. As shown in the figure, an overly small measurement noise can lead to overfitting to instantaneous rewards, which hurts convergence and final performance.

### 4.3.3 CASE STUDY

We also provide a case study from OpenMath data as Tab. 2 to show the improved reasoning ability of the KRPO model over the GRPO baseline. From the case, we observe that GRPO can figure out the prior steps, e.g. "This implies $4x = 3$, so $x = \frac{3}{4}$", but it stops here and derives the wrong answer 343. In contrast, the proposed KRPO can get the correct answer 1. For this question, the KRPO can get 12 corrects out of 12 trials, while the GRPO can have 3 corrects out of 12. This is because KRPO can better award and punish correct and incorrect answers via a more accurate advantage estimation.

## 5 RELATED WORK

### 5.1 DEEP REINFORCEMENT LEARNING

Reinforcement learning (RL) algorithms do not require strong supervision signals as supervised learning does, and they open another door for surpassing human performance. RL methods can be broadly divided into model-based and model-free methods. Despite a higher sample complexity, model-free methods remain dominant in many practical applications due to their relative simplicity and robustness.

Combining both policy (Mnih et al., 2013; Van Hasselt et al., 2016; Wang et al., 2016) and value (Sutton et al., 1999; Schulman et al., 2015; 2017) learning, actor-critic methods were proposed. This family includes Advantage Actor-Critic (A2C), which uses a critic to estimate the advantage function, and its asynchronous version Asynchronous Advantage Actor-Critic (A3C) (Mnih et al., 2016), which improves stability and exploration with multi-thread training. PPO can also be viewed as an actor-critic method with a clipped objective. In the continuous action setting, the Deep Deterministic Policy Gradient (DDPG) (Lillicrap et al., 2015) learns a deterministic actor and a critic. Twin Delayed Deep Deterministic Policy Gradient (TD3) (Fujimoto et al., 2018) improves DDPG by addressing function overestimation and policy update delays. Soft Actor-Critic (SAC) (Haarnoja et al., 2018) introduces entropy regularization to encourage exploration and improve stability.

These Reinforcement Learning algorithms can perform well for action control, e.g., playing Atari games (Van Hasselt et al., 2016), MoJoCo (Todorov et al., 2012), or other different simulations (Wang et al., 2021; 2020). However, they are not specifically designed for large language models (LLMs). Reducing the variance of policy gradient estimates is important for achieving stable and sample-efficient training. One widely used method is to subtract a baseline from the ob-

served returns. This baseline typically represents the expected return over all possible actions given the current state. With this subtraction, the resulting action advantage estimate should align more closely with the expected return under the current policy. In commonly used algorithms like Actor-Critic and Proximal Policy Optimization, this baseline is usually obtained from an auxiliary value function model. Nevertheless, maintaining and training this additional model significantly increases computation, particularly when working with large-scale models.

## 5.2 Reinforcement Learning to Improve Reasoning for LLMs

Learning a reinforcement learning model from human feedback has been proposed by Christiano et al. (Christiano et al., 2017) back in 2017. Recent advances in aligning large language models (LLMs) with human preferences have predominantly utilized reinforcement learning from human feedback (RLHF) (Ouyang et al., 2022). RLHF typically involves training a reward model based on human-labeled preferences and subsequently fine-tuning the LLM using reinforcement learning algorithms such as Proximal Policy Optimization (PPO). This approach has been instrumental in enhancing models such as ChatGPT (Achiam et al., 2023) and LLaMA (Touvron et al., 2023).

However, RLHF presents challenges, including the complexity of reward model training and the instability of reinforcement learning procedures. Alternative methods have been proposed to address these issues. Direct Preference Optimization (DPO) (Rafailov et al., 2023) eliminates the need for an explicit reward model by directly optimizing the policy to align with human preferences using a classification loss. This simplification leads to more stable and efficient training. Kahneman-Tversky Optimization (KTO) (Ethayarajh et al., 2024) incorporates principles from prospect theory to better model human decision-making biases. By aligning the training objective with human utility functions, KTO aims to improve the alignment of LLM outputs with human preferences. Reinforced Self-Training (ReST) (Gulcehre et al., 2023) adopts an offline reinforcement learning approach, generating training data from the model's own outputs and refining the policy without the need for online interaction or extensive human feedback. Rank Responses with Human Feedback (RRHF) (Yuan et al., 2023) simplifies the alignment process by ranking multiple model-generated responses and optimizing the model to prefer higher-ranked outputs, reducing the dependency on complex reward models. Reinforcement Learning from AI Feedback (RLAIF) (Lee et al., 2023) replaces human feedback with evaluations from a separate LLM, enabling scalable alignment without the need for human-labeled data. Self-Play Fine-Tuning (SPIN) (Chen et al., 2024) leverages self-play mechanisms, allowing the model to generate and learn from its own data, progressively improving performance without additional human supervision. Group Relative Policy Optimization (GRPO) (Shao et al., 2024) introduces a group-based approach to policy optimization, where the model's outputs are evaluated relative to a group baseline.

These methods represent a focus towards more efficient and scalable approaches for aligning LLMs with human preferences, reducing dependence on extensive human feedback and complex reinforcement learning procedures. But there is limited exploration from the estimation of a more accurate advantage. Therefore, in this paper, we propose Kalman Filter Enhanced Group Relative Policy Optimization (KRPO), which replaces the group-mean-based advantage estimation in GRPO with a Kalman-filter-based one. By treating observed rewards as noisy measurements of a latent reward signal, the filter adaptively estimates a baseline that tracks the underlying uncertainties. With a non-parametric one-dimensional Kalman filter, we can have more accurate advantage estimation by filtering out the underlying reward baseline and uncertainty with minimal computational cost.

## 6 Conclusion

In this work, we improve the model performance over GRPO by introducing KRPO, a Kalman filter-based method for accurate advantage estimation. Our approach replaces the static mean with a Kalman-filtered baseline and uncertainty that adapt to latent reward dynamics. KRPO maintains the simplicity of GRPO and requires no modification to the policy network or additional learning parameters. Empirical results show that KRPO improves training convergence and policy performance across environments with dynamic and noisy rewards. Besides math reasoning tasks, we are also interested in evaluating KRPO on other tasks such as creative writing and summarization. We consider this to be an important direction for future work.

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

## A  DATASETS

**Arithmetic Dataset** (Open-Thought, 2024). It contains 100,000 arithmetic problems involving addition, subtraction, multiplication, and division. Each problem is represented as a natural language prompt and the target is the correct answer as a string. The arithmetic expressions include up to 6-digit numbers and up to 6 terms. We first split the 100,000 problems into a training set and a testing set using the standard 80%/20% split. During both training and testing, we filter out problems

whose natural language prompt exceeds 128 characters. Then, we categorize the problems into three sub-datasets based on the number of digits and terms: the *easy* set includes problems with at most 3-digit numbers and at most 3 terms; the *normal* set includes problems with at most 5-digit numbers and at most 5 terms; and the *hard* set includes problems with at most 6-digit numbers and at most 6 terms.

**OpenMath-Instruct Dataset** (Toshniwal et al., 2024). This dataset contains 7,500 math problems from various topics, including Algebra, Geometry, Prealgebra, Precalculus, Number Theory, Intermediate Algebra, and Counting & Probability. Each problem is labeled with one of the five difficulty levels provided by the dataset. We split this dataset into a training set and a testing set using the same 80%/20% ratio. As with the first dataset, we filter out problems whose question text is longer than 128 characters during training and testing. We then group the problems into three sub-datasets based on difficulty level: the *easy* set includes problems with level $\leq 2$; the *normal* set includes problems with level $\leq 3$; and the *hard* set includes problems with level $\leq 5$.

**MATH500** (Lightman et al., 2023). MATH-500 is a test dataset of 500 math problems drawn from secondary/high school mathematics. Each instance consists of four fields: (1) problem, a mathematical question expressed in English with standard notation; (2) solution, a worked-out solution to the problem; (3) answer, the concise final answer; and (4) metadata including subject area (e.g. algebra, precalculus, geometry), difficulty level (integer from 1 to 5), and a unique identifier. Text fields (problem and solution) have average lengths in the thousands of characters; the answer field is very short. We also keep the standard 80%/20% split ratio for training and testing.

**AIME** (Veeraboina, 2024). AIME Problem Set (1983-2024) consists of 933 problems drawn from the American Invitational Mathematics Examination (AIME), covering the years 1983 through 2024. Each AIME contest contains 15 questions, and for each problem, the dataset provides the year, the problem number, the full problem statement, and the correct numerical answer. Problems are of high difficulty (designed for strong high school students), requiring non-trivial mathematical insight, covering a variety of topics such as algebra, number theory, combinatorics, geometry, etc. We divided the dataset into training and testing subsets with an 80%/20% split.

## B    MODEL PERFORMANCE ON MORE BASE-MODELS

Model Performance on additional base-models on Arithmetic-normal testing set are shown in Tab. 3, including Qwen2.5-0.5B-Inst and Qwen2.5-1.5B-Inst. The proposed KRPO outperforms GRPO and base models consistently and by a large margin.

To evaluate the generalization with larger models, we also conducted experiments with Llama3.2-3B-Instruct model as shown in the Fig. 5. With the Llama3.2-3B-Instruct model, the training returns also demonstrates that our proposed KRPO method can collect more rewards than the GRPO baseline, confirming the effectiveness of our approach.

## C    MODEL PERFORMANCE ON MORE DATASETS

Model Performance on additional MATH500 and AIME testing sets with Llama3.2-1B-Instruct as the base-model is shown in Tab. 4. Similar to the performance with different base models, the proposed KRPO outperforms GRPO and base models consistently.

Table 3: Model accuracy comparison between the KRPO and baseline models on Arithmetic-normal dataset. The best performance for each block with the same setting is **bolded**.

| Qwen2.5-0.5B-Inst | | Qwen2.5-1.5B-Inst | |
|---|---|---|---|
| **Model** | **Accuracy** | **Model** | **Accuracy** |
| Qwen2.5-0.5B-Inst | 8.042% | Qwen2.5-1.5B-Inst | 17.773% |
| GRPO-0.5B | 52.904% | GRPO-1.5B | 69.042% |
| KRPO-0.5B (Ours) | **68.825%** | KRPO-1.5B (Ours) | **88.510%** |
| Improvement | **+15.921%** | Improvement | **+19.468%** |

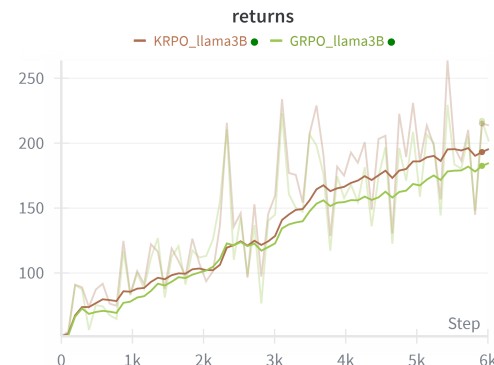

Figure 5: The training return curves of Llama3.2-3B-Instruct model on normal level of Arithmetic data.

Table 4: Model accuracy comparison between the KRPO and baseline models on MATH500 and AIME datasets with Llama3.2-1B-Instruct as the base-model. The best performance for each block with the same setting is **bolded**.

| MATH500 | | AIME | |
|---|---|---|---|
| **Model** | **Accuracy** | **Model** | **Accuracy** |
| Llama3.2 | 3.133% | Llama3.2 | 1.464% |
| GRPO | 24.086% | GRPO | 9.786% |
| KRPO (Ours) | **30.164%** | KRPO (Ours) | **12.336%** |
| Improvement | **+6.087%** | Improvement | **+2.550%** |

# D    MORE COMPARISONS WITH OTHER MODELS

Table 5: Model accuracy comparison between the KRPO and other models on the Arithmetic-hard dataset. The best performance for each block with the same setting is **bolded**.

| Model | Accuracy |
|---|---|
| Llama3.2 (Touvron et al., 2023) | 5.076% |
| DPO (Rafailov et al., 2023) | 10.220% |
| PPO (Schulman et al., 2017) | 17.650% |
| GRPO-fixed-b0 | 11.901% |
| GRPO-fixed-b3 | 0.349% |
| GRPO-fixed-b6 | 0.000% |
| GRPO (Shao et al., 2024) | 18.599% |
| KRPO (Ours) | **20.881%** |
| Improvement | **+2.282%** |

With respect to the baseline comparison, we formulated the Arithmetic-hard data into preferred and unpreferred pairs to enable optimization with Direct Preference Optimization (DPO) (Rafailov et al., 2023). The experimental results shown in the Tab. 5 indicate that KRPO remains the best-performing method for this task. This is due to its improved reward baseline and uncertainty estimation, which leads to more reliable advantage signals during optimization.

To have a more comprehensive comparison, we also conduct experiments to test the actual performance of fixed baselines. We found that the "fixed value" baseline raises a practical problem of what value to use. We compute the advantages within each group following the GRPO setup. As noted in the implementation details, we use a group size of 12, which means the rewards range from 0 to 12. Therefore, we chose 0 (without considering baseline), 6 (the midpoint) and 3 (the first quartile) as fixed-value baselines. If set to 0, the algorithm degenerates into a vanilla policy gradient without considering advantages. We tested this case and found that while learning is still

possible (the accuracy is only 11.901% as shown in the table), the optimization is less stable. If we set the baseline to 3 or 6, the optimization collapses entirely (approx. 0.000% accuracy). In the table, "GRPO-fixed-b0" denotes using 0 as the fixed baseline value, while "GRPO-fixed-b3" and "GRPO-fixed-b6" represent using 3 and 6 as fixed baseline values, respectively. This aligns with the understanding in the policy gradient literature of Sutton et al. (Sutton et al., 1999), where an effective baseline should approximate the expected return conditioned on the current policy. A fixed baseline cannot adapt to varying expected returns across states or tasks, resulting in it cannot fulfill the purpose of reducing the variance of policy gradient. So, such an approach is generally unsuitable for practical applications in most reinforcement learning settings.

## E  MODEL PERFORMANCE WITH DIFFERENT GROUP SIZE

An interesting point to be noted: the rollout size may influence the variance reduction behavior of both GRPO and KRPO. Nevertheless, this effect is not unique to KRPO. In GRPO, the group mean baseline also becomes less accurate when the number of rollouts per prompt is small, which is a natural consequence of the Law of Large Numbers via Monte Carlo sampling. The main difference being that KRPO provides an adaptive and stable advantage estimation within each group.

Therefore, we also conducted experiments on model performance with different group sizes (= {5, 8, 10, 15}) as shown in Fig. 6a. In the figure, "gx" denotes with group size x (e.g. g5 means with group size 5). Across various group size settings, the proposed KRPO consistently outperformed the baseline GRPO, demonstrating its robustness and effectiveness. Interestingly, as the group size varies from 5 to 15, the performance of both KRPO and GRPO improves; however, a large performance gap always remains between them, with KRPO maintaining a clear advantage.

## F  MODEL PERFORMANCE WITH DIFFERENT RANDOM SEEDS

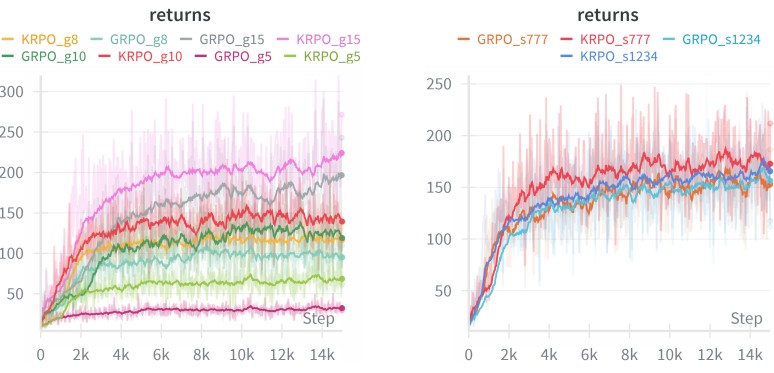

(a) Model performance with different group sizes.

(b) Model performance with different random seeds.

Figure 6: The training return curves of different (a) group sizes and (b) seeds. The base model is Llama-3.2-1B-Instruct on normal level of Arithmetic data.

We conducted experiments under multiple random seed settings to verify the robustness of the results, as shown in Fig. 6b. In the figure, "sx" presents with seed x (e.g. s777 means with random seed 777). We observe that when the random seeds are set to 777 and 1234, similar to the default seed 42, KRPO yields consistently higher collected rewards than GRPO, indicating that the superiority of KRPO is not sensitive to the choice of random initialization.

## G  TIME EFFICIENCY

We also present a comparison of the per-question training time between GRPO and the proposed KRPO. The Kalman Filter is a non-parametric method. The training time is measured in seconds.

According to the results, GRPO requires 2.271±0.5 seconds to complete training, while KRPO takes 2.288±0.5 seconds. The difference in training time is minimal, indicating that the introduction of the Kalman filter for advantage estimation in KRPO does not introduce computational overhead compared to GRPO. This suggests that the improved estimation in KRPO is achieved with nearly the same training efficiency. For the testing time, the Kalman filter will not be part of the evaluation process so no time burden will be added by it.

## H  MORE EXAMPLES OF CASE STUDY

Table 6: More case study of KRPO model question answering.

| Dataset | Data Sample | Thinking Process |
|---|---|---|
| Arithmetic | {"question": "What is -( -985 - -604 )?", "answer": "381"} | The calculation process involves: -( (-985 - (-604) ) ) = -( (-985 + 604 ) ) = -( -381 ) = 381 |
| Arithmetic | {"question": "What is 9147 + -1895 + -449?", "answer": "6803"} | Calculating the expression: 9147 - 1895 = 7252, then 7252 - 449 = 6803 |
| Arithmetic | {"question": "( -74 * -25 ) + -34 = ?", "answer": "1816"} | First, -74 * -25 = 1850, then 1850 + -34 = 1816 |
| Arithmetic | {"question": "Solve -( 427 - 202 ) + 514", "answer": "289"} | Let's calculate this step by step: -( 427 - 202 ) + 514 = -( 225 ) + 514, -( 225 ) = -225, -225 + 514 = 289 |
| OpenMath | {"type": "Prealgebra", "question": "Add 81.76 to 34.587 and round your answer to the nearest tenth."} | To add 81.76 and 34.587, we first add the numbers: 81.76 + 34.587 = 116.347. Then, we round to the nearest tenth, which gives us 116.3. |
| OpenMath | {"type": "Prealgebra", "question": "What is the greatest common factor of 32 and 48", "expected_answer": "16"} | To find the greatest common factor (GCF) of 32 and 48, we need to list the factors of each number. The factors of 32 are: 1, 2, 4, 8, 16, 32. The factors of 48 are: 1, 2, 3, 4, 6, 8, 12, 16, 24, 48. The greatest common factor is the largest number that appears in both lists, which is 16. |
| OpenMath | {"type": "Counting & Probability", "question": "Compute $\binom{7}{4}$.", "expected_answer": "35"} | To compute $\binom{7}{4}$, we use the formula for combinations: $\binom{n}{r} = \frac{n!}{r!(n-r)!}$. In this case, $n = 7$ and $r = 4$. Plugging these values into the formula, we get: $\binom{7}{4} = \frac{7!}{4!(7-4)!} = \frac{7!}{4!3!} = \frac{7 \times 6 \times 5}{3 \times 2 \times 1} = 35.$ |

We show more examples of how KRPO conducting the thinking process on different questions in Tab. 6. These cases show the strong reasoning ability of KRPO.

## I  THE ANALYSIS OF LEADING STATEMENT FOR LANGUAGE MODEL REASONING

From Tab. 1 and Tab. 6 we find that compared with OpenMath-Instruct data, the Arithmetic data questions are not harder, but the model receives much lower results, especially in the hard arithmetic questions. So we hypothesize that the leading statement might be a reason, as there are leading statements in OpenMath-Instruct data, but not in Arithmetic data. So we add leading statements, "Please calculate the following expression: " to each arithmetic question.

Interestingly, by enhancing with the leading statement (denoted with KRPO*), on the hard question set, the KRPO performance has increased largely (from 20.881% of KRPO to 40.554% of KRPO*). However, for easy and normal question sets, there are improvements with leading statements, but not as obvious as on a hard question set: on an easy set, KRPO 71.764% V.S. KRPO* 72.478%;

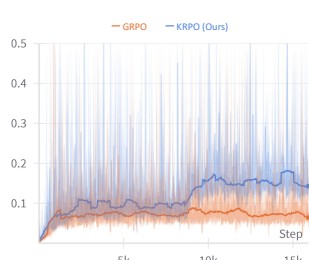 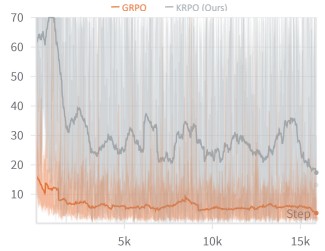 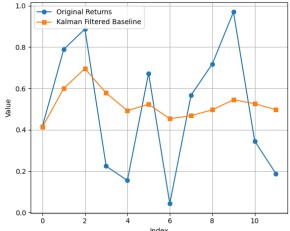

(a) KL divergence between the RL trained model and the reference model.

(b) Normalized gradients comparison between GRPO and the proposed KRPO.

(c) The Kalman filter gradually stabilizes the returns to a certain level.

Figure 7: The curves of (a) KL divergence; (b) normalized gradients of GRPO and the proposed KRPO; and (c) the Kalman filter gradually stabilizes the reward baseline to a certain level.

on the normal set, KRPO 50.589% V.S. 51.403%. It may be caused by the learning saturation of the model on easy and normal question sets as the questions have fewer terms and digits that are not that difficult to handle; but for the hard question set with a lot more terms and digits, the model will easily get lost in the understanding of just arithmetic questions, so some leading words can help greatly.

## J KL DIVERGENCE, NORMALIZED GRADIENTS AND REWARD BASELINE STABILIZATION ANALYSIS

We also examine the KL divergence of action probabilities between the RL finetuned model and its reference model in Fig. 7a. Interestingly, we notice that the KRPO has slightly higher KL divergence compared to the GRPO baseline. It makes sense as we observe from Tab. 1, the Llama3.2 model itself is not strong enough to perform ideally on the reasoning tasks, so deviating from the reference model but not too much would give the model more freedom to explore and fetch higher rewards.

For the gradients as Fig. 7, we applied gradient normalization and clipping to stabilize training by limiting the total gradient norm to a fixed $\ell_2$-norm threshold for the whole model. But the visualized gradient is before clipping. We can see that the KRPO has a higher gradient than the GRPO and gradually goes down, as the estimated variance from the Kalman Filter process can highlight the benefits/drawbacks of actions. The optimization signal is stronger and more informative, making it easier to optimize than its baseline model.

We provide an example to show the stabilization of the baseline from the Kalman Filter in Fig. 7c. With 12 rewards between [0,1], the Kalman filter gradually stabilizes the reward baseline.

