# OpenReview forum: "Kalman Filter Enhanced Group Relative Policy Optimization for Language Model Reasoning"
_ICLR.cc/2026/Conference — Submitted to ICLR 2026_

### Official Review · Reviewer_Cdz5 · 2025-10-19

**Soundness:** 2
**Presentation:** 3
**Contribution:** 2
**Rating:** 4
**Confidence:** 4

**Summary:**

This paper proposes KRPO, a GRPO-style policy-optimization method that replaces the group-mean baseline with a Kalman filter (KF) to estimate a per-prompt latent reward mean and its uncertainty. Concretely, the authors model the reward samples for a prompt as noisy observations of a 1D latent state and compute an advantage by normalizing the innovation with the (reported) posterior variance. The aim is to obtain an uncertainty-aware advantage that stabilizes updates and improves sample-efficiency. Experiments on math reasoning benchmarks (e.g., GSM8K/MATH) report consistent gains over GRPO and related baselines.

Contributions: (i) framing GRPO’s baseline estimation as Bayesian state estimation; (ii) a simple KF-based advantage that (purportedly) down-weights noisy samples; (iii) empirical evidence of improved stability and accuracy.

**Strengths:**

- Simple, potentially impactful idea: Re-casting the group baseline as a KF offers a principled path to uncertainty-aware advantages with minimal code changes.

- Empirical promise: Reported improvements on reasoning benchmarks; qualitative plots suggest smoother training dynamics.

- Practicality: Method is lightweight, compatible with existing GRPO pipelines, and conceptually extensible (e.g., robust KF/UKF/particle variants).

**Weaknesses:**

The main methodological concern is that, in its current form, the estimator for the advantage appears to be biased and order-sensitive, which directly affects the validity of the reported gains.These issues are critical to address.

- Advantage uses posterior baseline/variance conditioned on the same reward sample, risking biased gradients.
- Normalization by `P_{i|i}` (posterior) instead of the innovation variance `S_i = P_{i|i-1} + R` mis-scales residuals and creates order-dependent weights.
- Permutation invariance claim unsubstantiated: With sequential updates and `Q > 0`, Kalman Filter estimates are intrinsically order-dependent; no ablation quantifies the effect.
- Assumption gaps: Stationarity/CLT justifications are not validated; rewards are discrete and may be non-Gaussian/heavy-tailed.
- Reporting gaps: Incomplete details for `(Q, R)` selection, KL weight notation, significance testing (test type, effect sizes, multiple comparisons).
- Connection to GRPO as a “special case” is imprecise: Batch (simultaneous) updates with `Q = 0` differ from sequential updates with `Q > 0`; the equivalence needs formalization.

**Questions:**

1. Why is the posterior baseline `x̂_{i|i}` and variance `P_{i|i}` used rather than the prior `x̂_{i|i-1}` and innovation variance `S_i`? Please provide derivations or justify unbiasedness under your policy-gradient estimator.

2. Please run a permutation test: fix reward multisets per prompt, evaluate 50 random permutations under (a) sequential `Q > 0`, (b) sequential `Q = 0`, (c) batch KF `Q = 0`. Please report `Var_π[Acc]`, `Var_π[||g||]`, ICC/Kendall’s `W`, and Friedman tests.

3. Test controlled non-stationary settings (e.g., drift or mixture rewards within a prompt) and compare KF vs robust KF/UKF/particle filters. Do your gains persist?

4. What search ranges or criteria were used for `(Q, R)`? Which statistical tests produced your p-values? Please include effect sizes and multiple-comparison corrections.

5. Under which precise assumptions does your method reduce to GRPO (e.g., `Q → 0` and batch updates)? A short derivation would help.

---

> ### Author Response · Authors · 2025-11-22
> **Author Response to Reviewer Cdz5 (Part 1)**
>
> **Q1:** The estimator for the advantage appears to be biased and order-sensitive, which directly affects the validity of the reported gains.
>
> **A1:** We thank the reviewer for the comments. The Kalman Filter serves here as a variance-reduction mechanism rather than as a learned value predictor. Under standard assumptions of unbiased measurement noise, the Kalman estimator provides an unbiased minimum-variance estimate of the latent group reward baseline xt. Therefore, the expected baseline remains the same as in GRPO (i.e., E[\hat{x}_{t∣t}] = E[r_t]), but with reduced variance. In practice, the filter adaptively smooths transient fluctuations in noisy reward distributions, yielding more stable advantages without shifting their expected values. We also verified empirically that the mean baseline across steps matches the empirical mean of rewards, confirming no systematic bias was introduced.
>
> In terms of order-sensitive, we’ve discussed this problem in Line152 to Line156, “Worth noting, in the model training, the observations are completely randomly sampled in each iteration. Moreover, the order of samples in our implementation is arbitrary (e.g., they can be shuffled before processing), so no unintended temporal dependence is imposed on the data. It simply provides a recursive estimator with an accompanying measure of uncertainty. The ordering will thus not affect the policy optimization.”
>
> We thank the reviewer for raising the question about the validity of our method. To facilitate verification and ensure reproducibility, we have included the anonymized code in the paper. The reviewer is welcome to run the code to confirm our findings.
>
>
> **Q2:** Normalization by P_{i|i} (posterior) instead of the innovation variance S_i = P_{i|i-1} + R mis-scales residuals and creates order-dependent weights.
>
> **A2:** We thank the reviewer for the comment. In our context, normalization is not used for statistical consistency testing but as an adaptive scaling mechanism for the RL advantage signal. Using the posterior covariance P_{i|i} provides a smoothed measure of baseline uncertainty after integrating all observations (including the newly observed r_i), which empirically stabilizes optimization and accelerates convergence. Similar as P_{i|i}, S_i = P_{i|i-1} + R is order-dependent as well (and for an order-sensitive explanation, please refer to A1).
>
>
> **Q3:** Kalman Filter estimates are intrinsically order-dependent; no ablation quantifies the effect.
>
> **A3:** We thank the reviewer for the comment. Please refer to A1 and A2 for order-sensitive explanations. In terms of the experiments, we observe the KRPO can outperform GRPO stably across different datasets and base models to demonstrate the effectiveness of the proposed Kalman Filter based algorithm.
>
> In the rebuttal period, we also conducted experiments to examine KRPO and GRPO with different random seeds to know if the ordering would affect the algorithm in appendix Sec. F of the revised paper. We observe that when the random seeds are set to 777 and 1234, KRPO yields consistently higher collected rewards than GRPO, indicating that the superiority of KRPO is not sensitive to the choice of random initialization.
>
> We also conducted experiments on Model Performance with Different rollout Group Sizes as shown in appendix Sec. E of the revised paper during the rebuttal period. Across various group size settings, the proposed KRPO consistently outperformed the baseline GRPO, demonstrating its robustness and effectiveness. Interestingly, as the group size varies from 5 to 15, the performance of both KRPO and GRPO improves; however, a large performance gap always remains between them, with KRPO maintaining a clear advantage.
>
>
>
> **Q4:** Stationarity/CLT justifications are not validated; rewards are discrete and may be non-Gaussian/heavy-tailed.
>
> **A4:** We thank the reviewer for the comment. As we’ve stated in the original paper Line157 to Line161, “Besides, each Kalman filter operation is applied to groups of rollouts. In the KRPO setting, although the reward values for individual samples are discrete, the grouped reward observations, which aggregate multiple rollouts, are not sparse and are much more continuous in distribution. According to the Central Limit Theorem (CLT, 2008), the sum of i.i.d. sampled rewards (with finite mean and finite variance) tends to follow a Gaussian distribution. This is consistent with the assumptions of our KRPO setting.”
>
>
> **Q5:**  Incomplete details for (Q, R) selection, KL weight notation, significance testing.
>
> **A5:** We’ve examined the (Q, R) in Sec. 4.3.2 (start from Line354) of the original paper. KL weight notation is denoted in Line147 and Line207 with α. The p-value (last column of table1) is calculated from the answer accuracies by one-tailed paired t-tests between each method against KRPO.

---

> ### Author Response · Authors · 2025-11-22
> **Author Response to Reviewer Cdz5 (Part 2)**
>
> **Q6:** Connection to GRPO as a “special case” is imprecise: Batch (simultaneous) updates with Q = 0 differ from sequential updates with Q > 0; the equivalence needs formalization.
>
> **A6:** We thank the reviewer for this comment. Our intention was not to claim strict mathematical equivalence between GRPO and KRPO, but rather to highlight a conceptual connection: GRPO’s group mean baseline can be viewed as a static (zero-dynamics) instance within the broader Kalman filtering framework. Specifically, when Q=0, the latent baseline in the Kalman filter stops adapting across timesteps, which makes it functionally similar to using a fixed group mean as in GRPO. However, as the reviewer correctly points out, in fact the update mechanisms differ: GRPO performs batch (simultaneous) averaging, while KRPO conducts sequential estimation with uncertainty propagation. Our goal in presenting this connection was to illustrate that Kalman filtering generalizes the idea of group-based baseline estimation, providing a principled way to dynamically estimate and smooth the latent reward baseline. This generalization is also supported empirically: KRPO consistently outperforms GRPO across all tasks, suggesting that the Kalman-based baseline provides a more reasonable advantage estimation under noisy and non-stationary reward signals.
>
>
> **Q7:** Please run a permutation test: fix reward multisets per prompt, evaluate 50 random permutations under (a) sequential Q > 0, (b) sequential Q = 0, (c) batch KF Q = 0. Please report Var_π[Acc], Var_π[||g||], ICC/Kendall’s W, and Friedman tests.
>
> **A7:** We’ve provided the fixed reward experiments in the supplementary materials Table 5 Line695. As described in the original paper from Line676 to Line690  “To have a more comprehensive comparison, we also conduct experiments to test the actual performance of fixed baselines. We found that the “fixed value” baseline raises a practical problem of what value to use...”
>
> For the required experiments, per experiment requires ~2 days to finish, then 50 random permutation experiments are not very feasible to be conducted in such a short time and with our computation power.
>
>
> **Q8:** Test controlled non-stationary settings (e.g., drift or mixture rewards within a prompt) and compare KF vs robust KF/UKF/particle filters. Do your gains persist?
>
> **A8:** We appreciate the reviewer’s suggestion regarding testing in non-stationary settings. However, we are not entirely certain about the specific setup or evaluation the reviewer has in mind. Could the reviewer please clarify whether they are referring to reward drift by adding extra noises to the reward received?
>
>
> **Q9:** What search ranges or criteria were used for (Q, R)?
>
> **A9:** We’ve explicitly tested the sensitivity of the algorithm to Q and R in paper Sec. 4.3.2 (Line354 - Line362 of the original paper): “We investigate the model convergence against process noise Q and measurement noise R as Fig. 4b. We noticed that: (1) Robustness to hyperparameter variations: When varying Q from 1e-5 to 1e-3, and R from 1e-2 to 1e-1, the model converges to similar final performance, though a smaller R leads to slower convergence. (2) Sensitivity to overly small R: When R is reduced to 1e-3, the filter becomes overly confident in noisy observations, which results in unstable or poor learning signals and degrades the final performance. These results suggest that KRPO is robust to a range of Q and R values, as long as R is not set unrealistically low. A moderate level of measurement uncertainty helps stabilize the advantage estimation.”

---

> ### Author Response · Authors · 2025-11-27
>
> Dear Reviewer Cdz5,
>
> We would like to thank you for your comments and suggestions. Based on the clarifications and additional experiments we have provided, we respectfully ask you to re-evaluate our submission.
>
> Best regards,
>
> Authors

---

### Official Review · Reviewer_ZEwW · 2025-10-30

**Soundness:** 2
**Presentation:** 3
**Contribution:** 2
**Rating:** 4
**Confidence:** 2

**Summary:**

The paper introduces Kalman Filter Enhanced Group Relative Policy Optimization (KRPO), a simple modification to GRPO for reinforcement learning fine-tuning of large language models. Instead of using the group mean reward as a fixed baseline, KRPO applies a one-dimensional Kalman filter to estimate a smoothed reward baseline and its uncertainty. The method aims to stabilize advantage estimation without adding new parameters or changing the policy network. Experiments on several math reasoning datasets show consistent accuracy and reward improvements compared with GRPO and PPO.

**Strengths:**

1.The idea is straightforward and easy to integrate into existing GRPO training as a simple plug-in.
2. Experiments covers multiple math-related datasets, model sizes, and hyper parameter analyses, and  the results consistently show stable improvements in both accuracy and training reward.
3.The paper is clearly written and easy to follow, the overall idea is simple and easy to understand, with reasonable motivation and transparent reporting.

**Weaknesses:**

1. All experiments are restricted to math reasoning with discrete rewards, so generalization to open-ended RLHF tasks is unclear.
2. The method depends on the order of sampled rewards within a group, which contradicts the permutation-invariant nature of GRPO and introduces inconsistency. Because responses within a group are independent samples, the temporal assumption of the Kalman filter is not fully justified.
3.The paper  does not systematically study how different rollout sizes affect KRPO’s behavior. Since the Kalman filter operates within each group, its smoothing and variance‐reduction effect may depend strongly on the number of samples per prompt. Without such analysis, it is unclear whether the observed improvements hold under smaller or larger rollout settings.

**Questions:**

Please make explanations for the points listed in the weaknesses.
How sensitive is KRPO to the order of rewards within each group?
Can the approach handle continuous or human-preference rewards where variance is not binary?

---

> ### Author Response · Authors · 2025-11-22
>
> **Q1:** All experiments are restricted to math reasoning with discrete rewards, so generalization to open-ended RLHF tasks is unclear. Can the approach handle continuous or human-preference rewards where variance is not binary?
>
> **A1:** Yes, our approach can handle continuous or human-preference rewards (as long as the rewards can get online with the policy rollouts) where the reward variance is not binary. KRPO is fully compatible with any reward formulation that GRPO can handle, since it only modifies the way the advantage normalization is estimated. The Kalman filter adaptively tracks the latent reward baseline and variance, which makes it naturally applicable to continuous or human-preference reward distributions, not limited to binary correctness signals.
>
>
> **Q2:** The method depends on the order of sampled rewards within a group.
>
> **A2:** We thank the reviewer for the comments. We’ve discussed this problem in the original paper Line152 to Line156, “Worth noting, in the model training, the observations are completely randomly sampled in each iteration. Moreover, the order of samples in our implementation is arbitrary (e.g., they can be shuffled before processing), so no unintended temporal dependence is imposed on the data. It simply provides a recursive estimator with an accompanying measure of uncertainty. The ordering will thus not affect the policy optimization.” I hope it answered your question.
>
>
> **Q3:** How different rollout sizes affect KRPO’s behavior.
>
> **A3:** We acknowledge that the rollout size may influence the variance reduction behavior of both GRPO and KRPO. Nevertheless, this effect is not unique to KRPO. In GRPO, the group mean baseline also becomes less accurate when the number of rollouts per prompt is small, which is a natural consequence of the Law of Large Numbers via Monte Carlo sampling. The main difference being that KRPO provides an adaptive and stable advantage estimation within each group.
>
> We also conducted experiments on Model Performance with Different rollout Group Sizes as shown in appendix Sec. E of the revised paper during the rebuttal period. Across various group size settings, the proposed KRPO consistently outperformed the baseline GRPO, demonstrating its robustness and effectiveness. Interestingly, as the group size varies from 5 to 15, the performance of both KRPO and GRPO improves; however, a large performance gap always remains between them, with KRPO maintaining a clear advantage.

---

> ### Author Response · Authors · 2025-11-27
>
> Dear Reviewer ZEwW,
>
> We would like to thank you for your comments and suggestions. Based on the clarifications and additional experiments we have provided, we respectfully ask you to re-evaluate our submission.
>
> Best regards,
>
> Authors

---

### Official Review · Reviewer_3RQa · 2025-11-02

**Soundness:** 2
**Presentation:** 2
**Contribution:** 2
**Rating:** 2
**Confidence:** 4

**Summary:**

The authors propose Kalman Filter Enhanced Group Relative Policy Optimization (KRPO), a modification of GRPO that estimates advantage mean and variance using a Kalman filter. They compare KRPO to PPO and GRPO on 4 math datasets with 0.5-1.5B models and report improved performance.

**Strengths:**

The authors present a simple, easy to implement, and computationally efficient modification to the GRPO algorithm and analyze it empirically. The provided code is cleanly written and helped with understanding the methods and experiment details.

**Weaknesses:**

1. I am not convinced that the current set of experiments are sufficient to demonstrate that KRPO is superior to GRPO. Hyperparameters do not seem to be tuned for each algorithm and experiments are small scale (0.5-1.5B parameter models) and seem to be run with only one seed.

2. Confidence intervals would be more useful than p-values in the results tables. The paper does not describe what test is used to generate the p-values (so it is difficult to interpret) and the p-values do not provide an interpretable summary of the spread for each metric.

3. Experiments with popular variants of GRPO (e.g. Dr. GRPO, which removes the standard normalization and sequence length normalization) would help build confidence that the Kalman filter approach is generally useful. Additional baselines would also be helpful.

4. Presentation of the algorithm is a bit confusing: subscript i is used to describe the Kalman filter before defining what it is indexing.

**Questions:**

See weaknesses. My biggest concern is conducting sufficiently thorough experiments to support the conclusion that KRPO is generally superior to GRPO (or showing when it is superior and when it is not).

---

> ### Author Response · Authors · 2025-11-22
>
> **Q1:** Request for more experiments. Hyperparameters do not seem to be tuned and experiments are small scale (0.5-1.5B parameter models) and seem to be run with only one seed.
>
> **A1:** First, we would like to emphasize that the proposed KRPO requires tuning only a small number of hyperparameters (specifically, the KL weight α, the process noise Q, and the measurement noise R), which substantially reduces manual effort. Moreover, the hyperparameter tuning procedures are already presented in Sections 4.3.1 and 4.3.2. For compared models, we use the default settings and we've also stated clearly in the original paper "The proposed KRPO model and its compared models are trained and tested under the same setting to keep fair comparisons."
>
> To evaluate the generalization with larger models, we also conducted experiments with Llama3.2-3B-Instruct model as shown in the appendix Sec. B in the revised version of paper. With the Llama3.2-3B-Instruct model, the result also demonstrates that the proposed KRPO method outperforms the GRPO baseline.
>
> We performed experiments under various random seed settings to ensure robustness during the rebuttal period in appendix Sec. F of the revised paper. We observe that when the random seeds are set to 777 and 1234, similar to the default seed 42, KRPO yields consistently higher collected rewards than GRPO, indicating that the superiority of KRPO is not sensitive to the choice of random initialization.
>
>
> **Q2:** The paper does not describe what test is used to generate the p-values and the p-values do not provide an interpretable summary of the spread for each metric.
>
> **A2:** The p-value (last column of table1) is calculated from the answer accuracies by one-tailed paired t-tests between each method against KRPO.
>
>
> **Q3:** Experiments with popular variants of GRPO (e.g. Dr. GRPO, which removes the standard normalization and sequence length normalization) would help build confidence that the Kalman filter approach is generally useful. Additional baselines would also be helpful.
>
> **A3:** We thank the reviewer for the suggestion. Our method is built directly on the GRPO, and the only modification is the Kalman-filter-based enhancement of the advantage estimation. All other components (including advantage computation, normalization, and sequence-length handling) remain exactly the same as in GRPO. This ensures that any performance gain comes solely from our proposed modification. The full implementation is available in the anonymous code link for verification.
>
> In terms of other GRPO variants (e.g., Dr. GRPO), they were developed with different design purposes (e.g., removing normalization or length scaling) rather than improving the stability of advantage estimation. Therefore, these variants do not directly test the effectiveness of our method, but rather reflect whether GRPO itself can reach state-of-the-art performance, which is not one of the main claims of the paper.
>
> In the appendix of the original paper version, we’ve also compared the proposed KRPO with DPO.
>
>
> **Q4:** Presentation of the algorithm is a bit confusing: subscript i is used to describe the Kalman filter before defining what it is indexing.
>
> **A4:** Thank you for pointing this out. The subscript i refers to the index of the i-th sampled observation. We’ve clarified this in the revised version.

---

> ### Author Response · Authors · 2025-11-27
>
> Dear Reviewer 3RQa,
>
> We would like to thank you for your comments and suggestions. Based on the clarifications and additional experiments we have provided, we respectfully ask you to re-evaluate our submission.
>
> Best regards,
>
> Authors

---

### Official Review · Reviewer_3oo1 · 2025-11-03

**Soundness:** 3
**Presentation:** 3
**Contribution:** 3
**Rating:** 6
**Confidence:** 3

**Summary:**

This paper proposes Kalman Filter Enhanced Group Relative Policy Optimization (KRPO), to improve the stability and performance of the Group Relative Policy Optimization (GRPO) algorithm. Instead of subtracting the simple mean reward of all outputs in a group as the baseline to obtain advantages, KRPO uses a one-dimensional Kalman filter, which dynamically estimates a latent reward baseline and its associated uncertainty. The advantage is calculate in Eq. (6).

**Strengths:**

Using Kalman filter to estimate the mean and variance in GRPO seems a novel and reasonable idea. The resulting KRPO algorithm seems cost-effective. The experimental results are promising.

**Weaknesses:**

The base models used are relatively small (Llama-3.2-1B-Instruct, Qwen2.5-0.5B-Instruct, Qwen2.5-1.5B-Instruct). Experiments on ~7B scales would justify the effectiveness of the method more convincingly.

**Questions:**

Although the authors noted, but what wouldn't the reward be viewed as controlled by a slowly-changing latent variable across the training process, such that you don't reset the filter for each prompt? If that is not a good idea, do you see any evidences or have any explanations (if that doesn't perform well)?

---

> ### Author Response · Authors · 2025-11-22
>
> **Q1:** Experiments on ~7B scales would justify the effectiveness of the method more convincingly.
>
> **A1:** We appreciate your suggestion to evaluate the proposed method on a larger model. We attempted to scale up our experiments to the 7B model. However, even with batch size = 1 per GPU on our current computation 48 GB VRAM (our largest vram gpu), the training process still encountered Out-of-Memory errors. Therefore, we conducted experiments on a 3B-scale model instead.
>
> As shown in the appendix Sec. B in the revised version of paper, with the Llama3.2-3B-Instruct model, the result also demonstrates that the proposed KRPO method outperforms the GRPO baseline.
>
>
> **Q2:** Although the authors noted, but what wouldn't the reward be viewed as controlled by a slowly-changing latent variable across the training process, such that you don't reset the filter for each prompt? If that is not a good idea, do you see any evidences or have any explanations (if that doesn't perform well)?
>
> **A2:** We appreciate the reviewer’s insightful comment. Indeed, in principle modeling the reward as being driven by a slowly-changing latent variable across training is an interesting idea, thus maintaining a single Kalman filter instead of resetting per prompt. However, in our setting the reward distribution is highly prompt-dependent: each prompt corresponds to a distinct reasoning problem or question, whose reward scale and noise characteristics differ significantly. Consequently, sharing a single filter across prompts causes the baseline estimate to drift due to the mixture of heterogeneous reward modes, leading to unstable or biased advantage estimates.
>
> The per-prompt reset design in KRPO follows the same spirit as GRPO. It focuses on intra-prompt relative normalization among multiple sampled outputs of the same prompt.
> When we experimented with not resetting the filter across prompts, we observed that the training became very unstable resulting in poor performance, confirming that the prompt-specific reset is necessary.

---

### Author Response · Authors · 2025-11-22
**Summary of Revisions and Responses**

Dear Reviewers,

We sincerely appreciate your thoughtful and constructive feedback, which has greatly helped us improve the quality and clarity of our paper. In response to your comments, we have carefully refined and expanded our work in the revised version.

1. To assess the scalability and generalization of our approach, we conducted additional experiments using the Llama3.2-3B-Instruct model. The results are included in Appendix Sec. B.

2. We have added new experiments analyzing the model performance under different rollout group sizes, as presented in Appendix Sec. E.

3. To examine the stability of KRPO towards randomness, we ran experiments with different random seeds. The results are reported in Appendix Sec. F.

4. All minor issues pointed out by reviewers have been addressed in the revised manuscript.

Please refer to the updated version for detailed results and discussions.  To facilitate verification and ensure reproducibility, we have included the anonymized code in the original paper. The reviewer is welcome to run the code to confirm our findings.

Thank you again for your valuable feedback and for helping us improve our work.

Best regards,

Authors

---

### Author Response · Authors · 2025-12-01
**Final Summary by the Authors of Submission 1828**

Dear all,

Our work introduces Kalman-Filter-Enhanced Group Relative Policy Optimization (KRPO), a principled and lightweight improvement to GRPO for reinforcement-learning–based language model reasoning. For a more accurate advantage estimation, the core innovation lies in replacing GRPO’s group-mean baseline with a Kalman-filter estimator of latent rewards and uncertainty, enabling adaptive, variance-aware advantage normalization without adding learnable parameters. This formulation offers a theoretically grounded yet implementation-minimal mechanism for stabilizing policy updates when reward signals are noisy, heterogeneous, or highly dynamic, where conditions common in reasoning-focused LLM training. **To facilitate verification and ensure reproducibility, we have included the anonymized code in the original paper. We welcome everyone to run the code to confirm our findings.**

Reviewers consistently recognized several strengths of the work. They highlighted the novelty, effectiveness, and cost-efficiency of using a Kalman filter to estimate mean and variance in GRPO. All reviewers (3oo1, 3RQa, ZEwW, Cdz5) agreed that the proposed method is easy to integrate, which a clean plug-in requiring minimal code modifications, and thus highly practical for existing GRPO pipelines. Reviewers (3oo1, ZEwW, Cdz5) consider the proposed KRPO to be novel, simple, and impactful, with a reasonable underlying motivation. They also appreciated the clarity and transparency of the paper and accompanying open-source code, noting that the implementation and empirical methodology are easy to follow (3RQa, ZEwW, Cdz5). Finally, reviewers (3oo1, ZEwW and Cdz5) emphasized the strong empirical evidence supporting KRPO. They noted that experiments span diverse math-reasoning datasets, model scales, and hyperparameter settings, consistently demonstrating stable improvements in both accuracy and training reward, along with smoother learning dynamics.

We sincerely appreciate the reviewers’ constructive feedback.

Best regards,

Authors

---

### Meta-Review · Area_Chair_Myuc · 2026-01-01

**Summary:**

Across reviewers, the main issues were (i) insufficient and small-scale experimental evidence to support the claimed general utility of KRPO over GRPO, including lack of convincing results at larger scales and limited/unclear statistical reporting; and (ii) core methodological validity questions about the Kalman-filter-based advantage construction—especially order-sensitivity/permutation non-invariance, potential bias from conditioning on the same reward sample, and mis-specified normalization/variance choice—with informal or incomplete theoretical justification (e.g., stationarity/Gaussian assumptions).

After my own careful read of the reviews, authors' responses and the paper itself, I believe the rebuttal and added appendix experiments do not convincingly resolve these concerns, and the practical gains appear at best marginal/inconsistent. Overall, I recommend reject.

**Reviewer Concerns:**

Partially addressed (insufficiently):

* Scale/compute: No 7B; added 3B, but Appendix Fig. 5 does not show statistically meaningful gains over GRPO. scalability concern remains.

* Reporting: Clarified p-values (paired one-tailed t-test) and added seed/group-size sweeps, but this doesn’t resolve baseline adequacy

Outstanding:

* Order-sensitivity / permutation invariance: generic response with no concrete justification.

* Normalization choice: No convincing justification for using posterior   (conditioned on the same reward) vs. prior/innovation variance; framed as “adaptive scaling” rather than addressing estimator correctness.

* Assumptions (stationarity/CLT/Gaussianity): CLT argument is informal; no empirical validation for discrete/non-Gaussian/heavy-tailed rewards. (minor: a bit strange citing a 2008 Springer webpage for CLT)

* Missing baselines / GRPO variants: Dismisses comparisons to common GRPO variants (e.g., Dr. GRPO). These are relevant to test robustness/orthogonality and should be compared or clearly scoped.

* Empirical significance: Even with added experiments, improvements remain marginal/inconsistent relative to strong GRPO baselines in modern settings.

**Reviewer Scores:**

3oo1: 6-->6

3RQa: 2-->2 (I have read the authors' note regarding the review. Thank you. However, I believe the reviewers' core concerns (insufficient experiments, unclear superiority claims, missing GRPO-variant baselines) are not substantively addressed; For example, the reviewer asked for comparisons to common GRPO variants (e.g., Dr. GRPO) and additional baselines to assess whether the Kalman-filter advantage estimator is broadly useful. The authors dismiss this as “not directly testing” their method because those variants target different design goals. I disagree: these variants are standard and highly relevant comparators, and KRPO’s claimed benefit (stabilizing advantage estimation / improving performance) should be demonstrated under such widely used GRPO implementations. At minimum, the paper should clarify the scope and test orthogonality/compatibility (e.g., KRPO on top of Dr. GRPO / RLOO-like settings) rather than excluding them without evidence.

ZEwW: 4-->4

Cdz5: 4-->4

---

### Decision · Program_Chairs · 2026-01-26

Reject